# Spermidine Prevents Ethanol and Lipopolysaccharide-Induced Hepatic Injury in Mice

**DOI:** 10.3390/molecules26061786

**Published:** 2021-03-22

**Authors:** Raghabendra Adhikari, Ruchi Shah, Karina Reyes-Gordillo, Jaime Arellanes-Robledo, Ying Cheng, Joseph Ibrahim, Pamela L. Tuma

**Affiliations:** 1Department of Biology, The Catholic University of America, Washington DC 20064, USA; adhikari@cua.edu; 2Lipid Research Laboratory, VA Medical Center, Washington, DC 20422, USA; rsx@gwmail.gwu.edu (R.S.); karrygor@gwu.edu (K.R.-G.); jarellanes@inmegen.gob.mx (J.A.-R.); yingcheng2004@gmail.com (Y.C.); jomoabib@yahoo.com (J.I.); 3Department of Biochemistry and Molecular Medicine, The George Washington University Medical Center, Washington, DC 20037, USA; 4Division of Microbiology, Office of Regulatory Science, Center for Food Safety and Applied Nutrition, U.S. Food and Drug Administration, College Park, MD 20740, USA; 5Laboratory of Hepatic Diseases; Catedras-CONACYT–National Institute of Genomic Medicine (INMEGEN), Mexico City 14610, Mexico

**Keywords:** alcoholic liver disease, hepatic stellate cell activation, fibrosis, oxidative stress, spermidine

## Abstract

To date, there is no effective treatment for alcoholic liver disease, despite its prevalence world-wide. Because alcohol consumption is associated with oxidative stress-induced liver injury and pro-inflammatory responses, naturally occurring antioxidants and/or anti-inflammatories may be potential therapeutics. Spermidine is an abundant, ubiquitous polyamine that has been found to display strong antioxidant and anti-inflammatory properties. To further investigate whether spermidine is an effective intervention for alcohol-induced liver disease, we examined its hepatoprotective properties using a two-hit, chronic ethanol and acute lipopolysaccharide (LPS)-induced mouse model of liver injury. We determined that spermidine administration prevented ethanol and LPS-induced increases in liver injury using plasma ALT as a readout. Furthermore, histological analysis of tissue from control and treated animals revealed that the pathology associated with ethanol and LPS treatment was prevented in mice additionally treated with spermidine. As predicted, spermidine also prevented ethanol and LPS-induced oxidative stress by decreasing the levels of both reactive oxygen species (ROS) and lipid peroxidation. We further determined that spermidine treatment prevented the nuclear translocation of nuclear factor κB (NFκB) by blocking the phosphorylation of the inhibitory protein, IκB, thereby preventing expression of pro-inflammatory cytokines. Finally, by measuring expression of known markers of hepatic stellate cell activation and monitoring collagen deposition, we observed that spermidine also prevented alcohol and LPS-induced hepatic fibrosis. Together, our results indicate that spermidine is an antioxidant thereby conferring anti-inflammatory and anti-fibrotic effects associated with alcoholic liver injury.

## 1. Introduction

Alcohol itself is not hepatotoxic, rather, the metabolites and byproducts of its metabolism promote injury [1]. Because the liver is the major site of alcohol metabolism, it is most susceptible to alcohol-induced injury. The disease progresses in stages of increasing severity and irreversibility [2]. The early and reversible stages of the disease are characterized by steatosis that can progress to hepatic inflammation and injury. The accumulated hepatic injury ultimately promotes the later stages of alcoholic liver disease, which are characterized by fibrosis and cirrhosis, both of which are largely irreversible and untreatable. Although chronic alcohol consumption is associated with disease progression, patient studies have indicated that consumption itself is not enough to promote the later and more severe stages of the disease. Rather, a “second hit” must occur for end-stage disease to be realized. One such second hit is the bacterial endotoxin, lipopolysaccharide (LPS), that enters the circulation of alcoholics from a leaky gut [3,4]. Studies from alcoholic patients have demonstrated that LPS levels are elevated in the circulation, which has been confirmed in animal models of alcohol-induced liver injury [5,6]. The emerging hypothesis is that LPS promotes pro-inflammatory responses in the liver ultimately leading to cell death and fibrosis [2,3,4,7]. The now accepted “two-hit” model of disease progression posits that the first hit is injury due to ethanol metabolism and the production of reactive metabolites and oxidative stress followed by LPS release that promotes the second hit of nuclear factor κ (NFκB)-mediated pro-inflammatory signaling and the subsequent expression of pro-inflammatory cytokines in Kupffer cells.

Hepatic fibrosis begins as a wound response that includes the deposition of extracellular matrix that ultimately promotes tissue scarring [4,8]. In general, the response to injury described above leads to the activation of hepatic stellate cells, the main fibrogenic cells of the liver. The activated stellate cells become highly proliferative, synthesize and deposit large amounts of extracellular matrix and adopt a myo-fibroblastic phenotype characterized by the expression of platelet-derived growth factor-β receptor (PDGF-βR), collagen 1, fibronectin, and de novo synthesis of α smooth muscle actin (αSMA) [4].

Unresolved fibrosis can progress to cirrhosis and perhaps death. Because transplant is the only treatment for end-stage liver disease, much effort is aimed at finding potential novel therapeutics. The examination of naturally occurring hepatoprotective compounds is one strategy to identify novel treatments. Among such compounds is spermidine and its related ornithine-derived family members, spermine and putrescine. In general, these polyamines are thought to play an important role in the stabilization of DNA and regulation of gene expression and thus, protein synthesis, which ultimately regulates signal transduction, cell proliferation and differentiation. They have also been identified as modulators of age-related diseases due to their capacity to mimic the beneficial effects of caloric restriction/fasting on aging itself and age-related diseases (e.g., diabetes, cardiovascular disease, cancer, and neurodegeneration) [9]. Because aging and age-related diseases are often characterized by common features such as genomic instability, epigenetic changes, impaired proteostasis, mitochondrial dysfunction and chronic inflammation (among other alterations), these mimetics have been applied to a whole host of other diseases that share at least some of these features. From these studies, spermidine has emerged as both an antioxidant and an anti-inflammatory agent [9].

Based on these findings, and because ethanol-induced liver injury is characterized by oxidative stress and inflammation, we tested whether spermidine protects against alcohol-induced liver injury using a two-hit ethanol/LPS mouse model. The model was adapted from the two-hit ethanol/LPS model described by Hoek and colleagues [10] and the chronic ethanol/binge model proposed by Gao and colleagues [11]. In particular, mice were chronically fed ethanol for four weeks followed by an acute “binge” of ethanol. Importantly, this regimen is more consistent with not only the drinking behavior of chronic alcoholics (chronic consumption followed by bouts of intense bingeing), but includes the two insults required for severe stages of liver disease [3,4,7,8]. The rationale is that the first hit (chronic administration) promotes formation of dangerous and reactive alcohol metabolites and oxidative stress, whereas the second hit mimics LPS release from the leaky gut, which promotes pro-inflammatory responses ultimately leading to fibrosis. Also importantly, this model promotes more severe injury than other widely-used models where little to no fibrosis is observed [11].

## 2. Results

### 2.1. Spermidine Prevents Ethanol and LPS-induced Liver Injury

To first confirm that ethanol and LPS led to liver injury in the model system we used, we assessed liver morphology in hematoxylin and eosin-stained tissue sections (Figure 1A–D). As shown in Figure 1B, sections from mice treated with ethanol alone displayed features of liver injury including lipid droplet accumulation (as identified by their characteristic morphology) indicative of hepatic steatosis. Liver sections from mice treated with both ethanol and LPS also displayed lipid droplets, but enhanced injury was also observed with apparent areas of necrosis and inflammatory cell infiltration (Figure 1C). To confirm the histological analysis, we measured plasma ALT levels, a known marker of liver injury. Although modest increases were observed in mice treated with ethanol only, the levels of ALT in mice receiving both ethanol and LPS were markedly and significantly (*p* < 0.001) increased by almost fourfold (Figure 1E). Remarkably, in mice additionally treated with spermidine, liver morphology was virtually indistinguishable from control; no lipid droplet or inflammatory infiltration was observed (compare Figure 1A and D). Similarly, plasma ALT levels in spermidine-treated mice were restored to control levels and significantly lower (*p* < 0.001) than in mice treated with ethanol and LPS (Figure 1E) Together, these results indicated that our two-hit ethanol/LPS model resulted in significant liver injury that was prevented by spermidine treatment.

### 2.2. Spermidine Prevents Ethanol and LPS-induced Oxidative Stress and Lipid Peroxidation

To further assess the protective and presumed antioxidant effects of spermidine treatment, we measured reactive oxygen species (ROS) levels in control and treated cells. Consistent with results from Figure 1 and as predicted, the levels of ROS in mice treated with ethanol alone were significantly (*p* < 0.001) enhanced to over eightfold above levels seen in control mice (Figure 2A). In mice additionally treated with LPS, ROS levels were even higher at ~15-fold over control (Figure 2A). Although mice treated with spermidine displayed increased ROS levels, they were significantly (*p* < 0.001) decreased relative to mice treated with both ethanol and LPS (Figure 2A). As another readout for oxidative stress, we also monitored lipid peroxidation by immunoblotting for protein adduct formation by the peroxidation product, 4-hydroxy-2-nonenal (4-HNE). As observed for ROS levels, 4-HNE adduct formation (on multiple proteins) in mice treated with ethanol alone or with LPS was enhanced (Figure 2B). Densitometric comparison of immunoreactive bands (normalized to total β-actin levels) revealed that adduct formation in mice fed ethanol only was significantly increased (*p* < 0.001) threefold above control values (Figure 2C). In mice additionally treated with LPS, 4-HNE adduct formation was even higher at sixfold over control levels. Interestingly, spermidine was more effective in preventing lipid peroxidation than ROS per se. As shown in Figure 2B,C the levels of 4-HNE adduct formation in spermidine-treated mice were even lower than that seen in control. Together these results indicated that spermidine was displaying strong antioxidant properties as predicted.

### 2.3. Spermidine Prevents Ethanol and LPS-induced Activation of Inflammatory Mediators

To explore the presumed anti-inflammatory effects of spermidine treatment, we examined features of ethanol and LPS-induced inflammation including activation of the pro-inflammatory signaling molecules (NFκB and IκB) and the expression of pro-inflammatory cytokines. Because the activation of NFκB and its translocation to thenucleus promotes expression of pro-inflammatory cytokines, we measured nuclear vs. cytosolic distributions of NFκB in control and treated mice by immunoblotting. As predicted, in mice treated with ethanol or ethanol and LPS, increased NFκB nuclear levels (Figure 3A) were observed with reciprocal decreases in cytosolic levels (Figure 3B). Quantitation of the relative distributions in the nucleus (after normalizing to Lamin B1 levels) (Figure 3A) or the cytosol (after normalizing to β-actin levels) (Figure 3B) confirmed these observations. As observed in Figure 1 and Figure 2, the effects were far more pronounced in mice treated with both ethanol and LPS (Figure 3A,B). In contrast, no enhanced NFκB nuclear translocation was observed in mice additionally treated with spermidine (Figure 3A,B). As for 4-HNE adduction, spermidine treatment led to even lower levels of nuclear NFκB expression than observed in control. To further confirm these results, we immunoblotted for phosphorylated IκB, another marker of the NFκB-mediated inflammatory response. We observed that the ethanol and LPS-induced phosphorylation of IκB when normalized to total IκB levels was also prevented by spermidine addition. As shown in Figure 3C, the ratio of phospho-IκB to total IκB was significantly (*p* < 0.05) lower in spermidine-treated mice than in mice treated with ethanol and LPS.

To further confirm that spermidine treatment prevented pro-inflammatory signaling, we determined the relative gene expression levels of tumor necrosis fact-α (TNF-α), interleukin -1β (IL-1β) and interleukin 6 (IL6) in control and treated mice using qRT-PCR. Modest increases in the expression of all three pro-inflammatory cytokines (~1.2 to 1.5-fold) were observed in mice treated with ethanol alone, whereas much more robust and significant (*p* < 0.05) increases were observed in ethanol and LPS-treated mice to levels three to fourfold over the levels observed in control (Figure 3D–F). Spermidine addition significantly (*p* < 0.05) prevented this increase for all three cytokines to levels even lower than that seen in control (Figure 3D–F), indicating that pro-inflammatory cytokine expression was indeed prevented, consistent with decreased NFκB nuclear translocation and decreased IkB phosphorylation.

### 2.4. Spermidine Prevents Ethanol and LPS-induced Hepatic Fibrosis

To determine whether spermidine addition was also anti-fibrogenic, we immunoblotted tissue lysates for known markers of hepatic stellate cell activation including α-SMA, PDGF-βR and fibronectin in control and treated mice. Densitometric comparison of immunoreactive bands (normalized to total β-actin levels) revealed that all three markers displayed enhanced expression in mice treated with ethanol alone (two to 3.5-fold over control) and in mice treated with ethanol and LPS (2.5 to fourfold over control) (*p* < 0.05) (Figure 4A–C). Mice exposed to spermidine expressed these fibrogenic proteins closer to control levels, indicating that fibrogenesis was prevented. To confirm these results, we stained tissue sections with Sirius red to monitor collagen deposition in control and treated mice. As shown in Figure 5B, ethanol exposure led to modest increases in collagen deposition indicated by small regions of Sirius red labeling. In contrast, large Sirius red positive regions were observed in mice treated with both ethanol and LPS indicative of enhanced fibrogenesis (Figure 5C). As observed for the other fibrogenic markers, spermidine treatment prevented enhanced collagen deposition, further confirming its anti-fibrogenic capacity. We also determined hydroxyproline content in tissues from control and treated mice as a final test for the extent of fibrogenesis. As for the other markers, hydroxyproline content was greatly (~fourfold) and significantly (*p* < 0.0001) increased in mice treated with ethanol and LPS, and spermidine treatment significantly (*p* < 0.0001) prevented this increase (Figure 5B). Together, these results indicated that spermidine was anti-fibrogenic in the two-hit model of alcohol/LPS-induced injury.

## 3. Discussion

Although polyamine levels have been shown to be decreased upon ethanol exposure [12], to our knowledge this is the first report to show that spermidine supplementation is protective against alcohol-induced liver disease. This is of particular importance given that there are no treatments for end-stage liver disease besides transplantation. The attractiveness of spermidine and the related polyamines and other caloric restriction mimetics is that they are highly abundant molecules in many dietary sources. Not only are they enriched in common foods like aged cheese, mushrooms, soy, legumes, corn, and whole grains, they are also especially enriched in more exotic foods like natto (a traditional Japanese dish made from fermented soybeans) and Durian fruit varieties (fruits from Southeast Asia known for their strong orders) [13]. The Mediterranean diet is also attributed with the highest levels of spermidine and has long been associated with healthy eating. Because ingested spermidine is easily adsorbed from the gut and is distributed without degradation [14], changes in diet can easily change spermidine concentrations in the circulation.

Although the specific mechanisms by which spermidine is thought to confer its beneficial effects are not known, the underlying broad mechanism of action of the caloric fasting mimetics is their apparent capacity to promote protective autophagy [15]. This is consistent with the findings that spermidine induces protective autophagy in liver and leads to extensive changes in the liver metabolome [16]. Although we have not yet examined whether autophagy is induced in our model system, the prediction is that it would be. In further support of our findings that spermidine is protective against liver injury are studies where oral administration of spermidine and spermine was found to ameliorate liver-ischemia/reperfusion injury and to promote liver regeneration in rats [17]. Furthermore, spermidine was protective against CCl_4_-induced liver fibrosis/cirrhosis and DEN-induced hepatocellular carcinoma in mice [18,19].

Based on these results, results from other tissues and the results presented herein, we predict that spermidine is acting as an antioxidant in our model system. Its administration relieves the oxidative stress associated with alcohol metabolism and prevents the ethanol/LPS-induced inflammatory response and stellate cell activation, which, in turn, ameliorates the subsequent progression to fibrosis. This prediction is also consistent with the demonstrated effectiveness of polyamines in protecting against enhanced mitochondrial permeability in rat liver [20,21]. Because spermidine administration was shown to protect the mucosal barrier in cultured intestinal epithelial cells and prevent LPS leakage [22], we further predict that such protection is occurring in our model system. Further studies are needed to establish whether this promising compound and related polyamines are indeed hepatoprotective and can serve as effective interventions for patients suffering from alcohol-related injury.

## 4. Materials and Methods

### 4.1. Mice and Diet

Eight-week-old female mice were purchased from Charles River (Wilmington, MA). The mice were housed in pairs in plastic cages in a temperature-controlled room at 25 °C with 12-h light–dark cycles. Pelleted commercial diet (Purina Rodent Chow, #500, TMI Nutrition, St. Louis, MO) was fed to all mice during the first week of acclimation after their arrival. The animal experiments were carried out in accordance to protocols approved by the Washington DC Veterans Affairs Medical Center Institutional Animal Care and Use Committee. Mice were separated into four groups of four mice each: (1) control; (2) ethanol-treated; (3) ethanol and LPS-treated; and (4) ethanol, LPS and spermidine-treated. The mice were pair-fed Lieber DeCarli control or ethanol liquid diets (36% total fat calories) with fish oil that is high in ω3 fatty acids (14.1% of calories as ω3 fatty acids) for four weeks. The isocaloric diets and their formulations were produced according to the modified method of Lieber and DeCarli [23] with the recommended normal nutrients, vitamins and minerals defined in the AIN-93 diet [24]. Thus, 36% of the total energy of the ethanol diet was from fat, 20% from protein, 36% from ethanol and the rest from carbohydrate. The corresponding isocaloric control diet had isoenergetic amounts of dextrin maltose in place of ethanol. The ethanol concentrations in the liquid diet were gradually increased starting at 1% on day one and adjusted to a final concentration of 5% over a seven-day period. The mice fed the ethanol diets also received a single dose of 5 g/kg body weight of ethanol by gavage, in the presence or absence of LPS (2 mg/kg body weight, i.p.), 6 h prior to euthanizing the animal. Spermidine (Sigma-Aldrich, Cat. No. S0266) was administered intraperitoneally as a daily dose of 1 mg/kg of body weight (diluted in distilled water) for 1 week prior to euthanizing the animal.

### 4.2. Plasma Liver Injury Markers

The levels of liver injury were assessed by measuring plasma alanine aminotransferase (ALT) with assays purchased from Teco Diagnostics (Anaheim, CA, USA) and performed according to the manufacturer’s instructions.

### 4.3. Hematoxylin and Eosin Staining

Liver tissues were fixed and processed for staining with hematoxylin and eosin using routine procedures as previously described [25].

### 4.4. Reactive Oxygen Species and Hydroxyproline Determinations

ROS levels were determined using the OxiSelect™ In Vitro ROS/RNS Assay Kit (Cat. No. STA-347, Cell Bio Labs, San Diego, CA, USA). Samples were added to 96-well plates with catalyst for 5 min at RT to accelerate the oxidative reaction. Dichlorodihydrofluorescein was added and the plates for 45 min. The resultant fluorescent product was measured fluorometrically and levels determined against an H_2_O_2_ standard.

Hydroxyproline content was measured colorimetrically using the Hydroxyproline Assay kit (Cat. No. MAK008-1KT, Sigma-Aldrich). Ten mg liver tissue was homogenized in 100 μL of H_2_O and transferred to a pressure-tight vial containing 200 μL of concentrated HCl. Activated charcoal was added to remove impurities. Samples were hydrolyzed at 120 °C for 48 h. Twenty-five μL of supernatant was dried by incubation in a 60 °C oven. One hundred μL of oxidation buffer was added to each well and incubated at RT for 5 min followed by the addition of 100 μL of 4-(Dimethylamino)benzaldehyde diluted with perchloric acid/isopropanol and incubated at 60 °C for 90 min. Absorption was measured at 560 nm.

### 4.5. Total, Nuclear and Cytosolic Protein Extraction

Total protein was extracted from liver tissue by homogenization in lysis buffer containing 1 mol/L Tris (pH 8), 5 mol/L NaCl, 0.5 mol/L EDTA, 0.5 mol/L NaF, 100 mmol/L sodium pyrophosphate, 100 mmol/L Na_3_VO4, 200 mmol/L PMSF. Nuclear and cytosolic fractions were prepared using kits purchased from Thermo Scientific (Rockford, IL, USA). The fractionations were performed according to the manufacturer’s instructions.

### 4.6. SDS-PAGE and Western Blot Analysis

Protein concentrations were determined using the bicinchoninic acid assay (Pierce Chemical, Rockford, IL, USA). Proteins from tissue homogenates were electrophoretically separated by SDS-PAGE, transferred to membranes and immunoblotted as previously described [26]. Proteins were detected with a NEN Life Science Products Renaissance enhanced chemiluminescence system (PerkinElmer, Waltham, MA) according to the manufacturer’s recommendations. Immunoblots were probed with antibodies specific to NFκB, inhibitor of NFκB (IκB) and PDGF-βR (purchased from Cell Signaling, Danvers, MA, USA), phospho-IκB (pIκB), αSMA and fibronectin (purchased from Abcam, Cambridge, MA, USA), and β-actin (purchased from Sigma-Aldrich). Relative protein levels were determined by densitometric analysis of immunoreactive bands and normalized to total β-actin levels (antibody purchased from Sigma-Aldrich) or Lamin B1 levels (antibody purchased from Abcam) for nuclear fractions.

### 4.7. RNA Extraction and Quantitative RT-PCR

RNA from liver tissue was extracted using TriZol reagent (Life Technologies, Carlsbad, CA, USA). cDNA templates were synthesized and quantitative RT-PCR was performed as previously described [26]. The 40S ribosomal protein S14 was used as the standard housekeeping gene. Fold-changes in gene expression were calculated using the double delta cycle threshold (CT) method. The delta CT was calculated between the gene of interest and the reference housekeeping gene. The double delta was calculated by taking the difference of the delta CT for the test group and the delta CT for the control group. Fold-changes were calculated using the 2^ (delta CT) formula.

### 4.8. Sirius Red Staining

To assess the degree of fibrosis, tissue sections were stained for collagen 1 and 3 fibers using Picro-Sirius red (Abcam). Briefly, thick liver sections (4–6 μm) were deparaffinized and incubated for 1 h with Picro-Sirius red. The sections were rinsed twice in acetic acid and dehydrated by dipping twice in absolute alcohol. Sections were mounted with Eukitt quick hardening mounting medium (Sigma-Aldrich) and visualized using a Zeiss 510 microscope (Carl Zeiss, Thornwood, NY, USA) equipped with a 20X objective. Five random fields from each animal group were visualized and relative staining of collagen was assessed in a blinded fashion. The percent of fibrosis was calculated based on the intensity of Sirius red staining using ImageJ (NIH, Bethesda, MD, USA).

### 4.9. Statistical Analysis

All experiments were performed in triplicate. Values are expressed as mean ± SE from at least three independent experiments. Comparisons were made between experimental groups using the Student’s two-tailed *t*-test for paired data. *p* values ≤ 0.05 were considered significant.

## Figures and Tables

**Figure 1 molecules-26-01786-f001:**
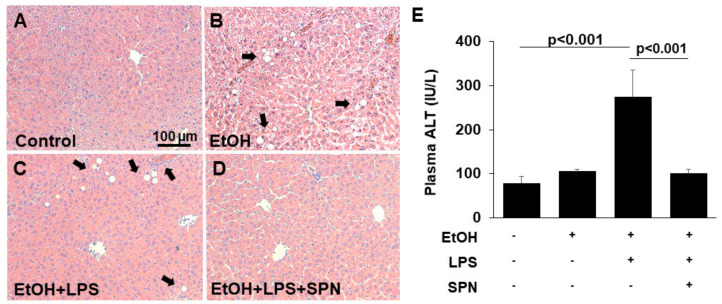
Spermidine prevents ethanol and LPS-induced liver injury. Thick (4–6 µm) sections from mouse livers treated in the absence (control) or presence of ethanol (EtOH), LPS and/or spermidine (SPN) as indicated were stained with hematoxylin and eosin using routine procedures (**A**–**D**). Arrows indicate lipid droplets (**B**,**C**) and areas of inflammatory cell infiltration (**C**) that are absent in sections from control (**A**) or spermidine-treated mice (**D**). Bar = 100 µm. (**E**) Plasma levels of ALT were determined in control or treated mice as indicated. For each experimental condition, four mice were anlayzed (*n* = 4). The values in E represent the means of triplicate experiments ± SE.

**Figure 2 molecules-26-01786-f002:**
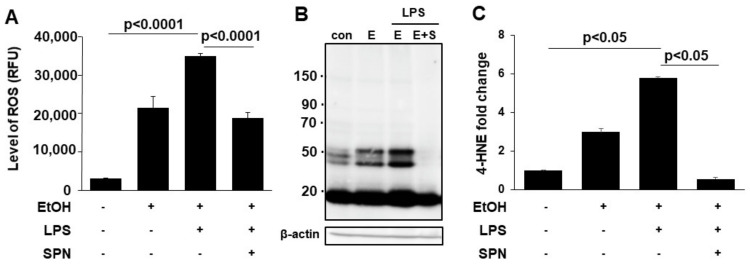
Spermidine prevents ethanol and LPS-induced oxidative stress. The levels of ROS were determined fluorometrically in livers from mice treated in the absence (control) or presence of ethanol (EtOH), LPS and/or spermidine (SPN) as indicated (**A**). Total 4-HNE protein adducts (**B**). Molecular weight markers are indicated on the left of the immunoblot in kDa. Densitometric comparison of the immunoreactive species was performed to determine relative levels of expression. Values were normalized to total β-actin levels and are plotted as fold-increase relative to control (**C**). A representative immunoblot is shown in (**B**). For each experimental condition, four mice were analyzed (*n* = 4). The values in (**A**,**C**) represent the means of triplicate experiments ± SE.

**Figure 3 molecules-26-01786-f003:**
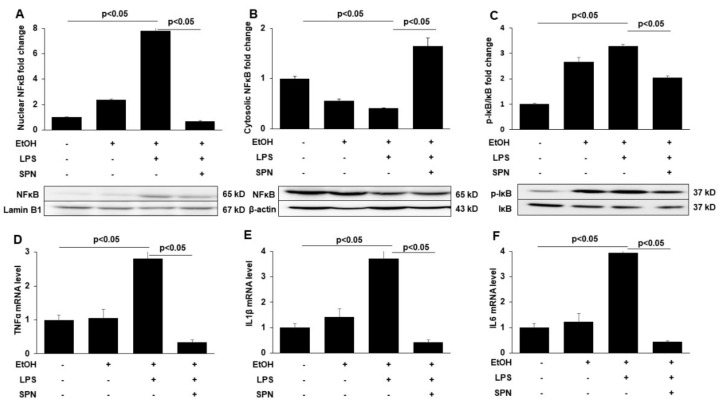
Spermidine prevents ethanol and LPS-induced NFκB nuclear translocation, IκB activation by phosphorylation and pro-inflammatory cytokine expression. Nuclear (**A**) and cytosolic (**B**) fractions were isolated from livers of mice treated in the absence (control) or presence of ethanol (EtOH), LPS and/or spermidine (SPN) as indicated and immunoblotted for NFκB. Representative immunoblots from both fractions are shown. Densitometric comparison analysis of the immunoreactive species was performed and normalized to Lamin B1 levels (for the nuclear fractions) or β-actin (for the cytosolic fractions). The relative distributions were determined and plotted as fold-increase over control. (**C**) Total protein isolated from control or treated mice was immunoblotted for total and phosphorylated IκB (pIκB). The ratio of phospho-IκB vs. total IκB was calculated from densitometric analysis of immunoreactive species. Representative immunoblots are shown. Values are plotted as fold-change. Total mRNA isolated from livers of control or treated mice was processed for qRT-PCR to determine expression levels of TNFα (**D**), IL1β (**E**) or IL6 (**F**). For each experimental condition, four mice were analyzed (*n* = 4). The values are means of triplicate experiments ± SE.

**Figure 4 molecules-26-01786-f004:**
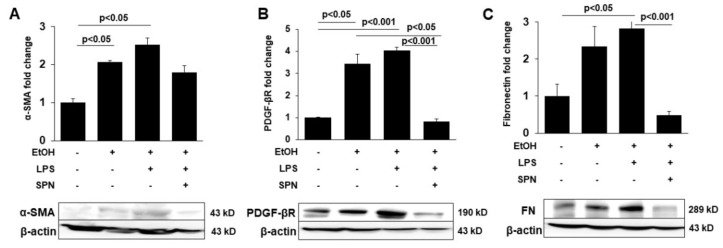
Spermidine prevents ethanol and LPS-induced expression of markers of hepatic stellate cell activation and fibrogenesis. Whole homogenates were prepared from livers of mice treated in the absence (control) or presence of ethanol (EtOH), LPS and/or spermidine (SPN) as indicated and immunoblotted with antibodies specific to αSMA (**A**), PDGF-βR (**B**) or fibronectin (FN) (**C**). Densitometric comparison of the immunoreactive species was performed to determine relative levels of expression. Values were normalized to total β-actin levels and are plotted as fold-increase. Representative immunoblots are shown for each marker. For each experimental condition, four mice were analyzed (*n* = 4). The values represent means of triplicate experiments ± SE.

**Figure 5 molecules-26-01786-f005:**
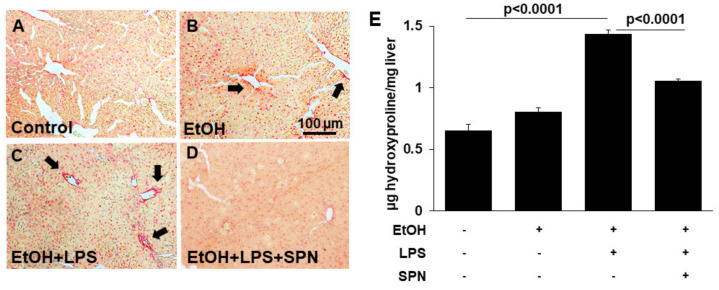
Spermidine prevents ethanol and and LPS-collagen deposition. Thick (4–6 µm) sections from mouse livers treated in the absence (control) or presence of ethanol (EtOH), LPS and/or spermidine (SPN) as indicated were stained with Sirius red to monitor collagen deposition (**A–D**). Labeling was only observed in sections from mice treated with ethanol alone or additionally treated with LPS (**B**,**C**). Bar = 100 µm Total hydroxyproline was detected colorimetrically from control or treated mice as an additional indicator of collagen deposition (**E**). For each experimental condition, four mice were analyzed (*n* = 4). Values are plotted as µg hydroxyproline detected per mg of liver. The values in E are means of triplicate experiments ± SE.

## Data Availability

Data are available from the corresponding author upon request.

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
