# Peer review of "Spermidine Prevents Ethanol and Lipopolysaccharide-Induced Hepatic Injury in Mice"

_molecules, 2021, doi:10.3390/molecules26061786_

Round 1
Reviewer 1 Report
The manuscript entitled “Spermidine prevents ethanol and lipopolysaccharide-induced hepatic injury in mice” by Adhikari et al. shows the role of spermidine as a hepatoprotective compound as well as its possible indication for preventing/mitigating the harmful effects of alcohol on the liver. The manuscript is well written, the experiments were adequate to the authors' objectives. However, the discussion needs to be better elaborated. I suggest that the authors compare their encounters with the literature explaining what was found that would support results already described or that contradict the same results. In the current version there is a repetition of the functions of spermidine and the methodology, already described in the introduction. Also, emphasize your meetings that are very important and unpublished.
Author Response
Reviewer 1
1. The manuscript is well written. The experiments were adequate to the authors' objectives.
However, the discussion needs to be better elaborated. I suggest that the authors compare
their results with the literature explaining what was found that would either support or
contradict results already described.
Upon rereading our Discussion based on this reviewer’s comment, we agree that our studies
were not put into context with other studies performed in liver. We have gone back to the
Discussion and rewrote it putting our results better in context.
2. In the current version there is a repetition of the functions of spermidine and the methodology
already described in the introduction.
We agree that the Introduction and Discussion as first written were redundant with respect to
spermidine functions and methods used. We have completely rewritten sections of the
Introduction and Discussion. We have added text, deleted text and moved text around and
to remove redundancy and to enhance the overall clarity of the information provided.
3. Emphasize why your results are very important and unpublished.
We have better emphasized how our results contribute to understanding the protective
capacity of spermidine with respect to ethanol/LPS-induced liver injury. Although ethanol
exposure has been shown to alter polyamine metabolism leading to decreases in spermidine,
putrescine and spermine, to our knowledge this is the first report that demonstrates that
spermidine supplementation is protective against ethanol/LPS-induced liver injury. We have
emphasized that in the first line of the Discussion. We have also emphasized the need for
treatments for alcoholic patients and that spermidine supplementation may be a viable option.
Reviewer 2 Report
In the manuscript, the authors examined the protective effect of spermidine in ethanol and lipopolysaccharide-induced hepatic injury in mice. They revealed that spermidine increased anti-oxidation protein expression and reduced liver injury significantly. Although the detailed molecular mechanism remains unclear, the data is clearly presented except for some minor comments that are discussed below.
Specific comments:
- The authors should state how to decide the route and dose of spermidine in the overall study. Why did the authors use one dose (1 mg/kg) and one route (i.p.)? Did the authors try several different ways? Otherwise, it may be appropriate to include references regarding the treatment of spermidine in mice.
- Figure 1: More direct evidences are needed to state that arrows indicate lipid droplets (e.g. Oil Red O staining). They might be vacuoles from hepatocyte degeneration.
- Figure 1 and 5: The authors should provide scale bars to all images.
- Figure 2B: The author should show the uncropped image of 4-HNE immunoblot because there should be several HNE-adducted proteins.
- Figure 3 (in legend): The description of A and B would need to be reversed.
- Figure 5D: The images are dark and not clear. The contrast is differently adjusted from others. Please replace it with more appropriate one.
- I recommend checking to see some apoptotic markers (e.g. the activated caspase-3).
- The authors should add the n (sample number) for each experiment in all of the figure legends.
- Materials and methods: The authors should describe the method for the evaluation of ROS and hydroxyproline content in detail (catalog number needed)
- Materials and methods: The authors should state how to dissolve spermidine for treating with mice.
Author Response
Reviewer 2
1. The authors should state how the route and dose of spermidine in the overall study was
decided. Why did the authors use one dose (1 mg/kg) and one route (i.p.)? Did the authors
try several different ways? Otherwise, it may be appropriate to include references regarding
the treatment of spermidine in mice.
Spermidine and related compounds are all soluble in water and can be delivered in a variety
of ways with the same results. It can be administered in drinking water for short or long periods
or orally via gastric gavage. Others have injected it iv or ip. The concentrations used widely
vary also depending on how the experiments are designed – to resemble uptake through diet
or taking a supplement (longer-term at lower concentrations) or an acute treatment (at higher
concentrations delivered in one or few doses). To ensure all animals received the same dose
of spermidine, we chose ip administration rather than an ad libitum feeding regimen. We also
chose a lower dose to better mimic diet supplementation. Further studies are planned that
will better examine a wider range of spermidine concentrations with oral administration to
better understand how it can be used as a dietary supplement in humans.
2. Figure 1: More direct evidence is needed to state that arrows indicate lipid droplets (e.g. Oil
Red O staining). They might be vacuoles from hepatocyte degeneration.
It is widely known and fully accepted that alcohol exposure is associated with lipid droplet
formation and accumulation (fatty liver) and that this represents the first (and reversible) stage
of alcohol-induced liver injury. Their presence is highly predicted and expected. In general,
droplet formation is confirmed by a pathologist based on their characteristic morphology in
hematoxylin and eosin stained images. Droplets appear as white (they are dye impermeant),
nearly perfectly round structures with an apparent limiting membrane. This morphology is
unlike that seen in degeneration where lesions are irregularly shaped and have no limiting
membrane. One of the first publications that described the two-hit LPS/ethanol model of liver
injury (almost 20 years ago) used this standard of assessing lipid droplet accumulation
(Koteish A et al, 2002. Chronic ethanol exposure potentiates lipopolysaccharide liver injury
despite inhibiting Jun N-terminal kinase and caspase 3 activation. J Biol Chem.
12;277(15):13037-44). In the alcohol-induced liver injury field, pathologist identification of lipid
droplets is an accepted practice.
3. Figure 1 and 5: The authors should provide scale bars to all images.
We apologize for the omission. Scale bars have been added to Figures 1 and 5 as requested.
4. Figure 2B: The author should show the uncropped image of 4-HNE immunoblot because there
should be several HNE-adducted proteins.
We have included the uncropped immunoblot as requested to show the many 4-HNEadducted
proteins.
5. Figure 3 (in legend): The description of A and B would need to be reversed.
Thank you for catching our mistake. We have corrected the legend.
6. Figure 5D: The images are dark and not clear. The contrast is differently adjusted from others.
Please replace it with more appropriate one.
We agree the contrast in the images in Figures 5 A-D were mismatched. We have done our
best to select matched images that are more visually appealing.
7. I recommend checking to see some apoptotic markers (e.g. the activated caspase-3).
One of the first publications that described the two-hit LPS/ethanol model of liver injury
(mentioned above) examined apoptosis directly and as the title indicates, no change in
caspase 3 activation was observed (Koteish A et al, 2002. Chronic ethanol exposure
potentiates lipopolysaccharide liver injury despite inhibiting Jun N-terminal kinase and
caspase 3 activation. J Biol Chem. 12;277(15):13037-44). This was also observed for
caspase 8 activation in the same publication.
8. The authors should add the n (sample number) for each experiment in all of the figure legends.
We have added the number of mice analyzed per group (n = 4) and have added that to the
figure legends. All experiments were done in triplicate. That is also included in the legends.
9. Materials and methods: The authors should describe the method for the evaluation of ROS
and hydroxyproline content in detail (catalog number needed).
As recommended, we have added the catalog numbers for each assay kit to the Methods
section and expanded on the methods themselves to more fully describe how the assays were
performed.
10. Materials and methods: The authors should state how to dissolve spermidine for treating with
mice.
The spermidine was dissolved in water. We have added that to the Methods section.
Round 2
Reviewer 2 Report
I have no further comments.